# SecrecyPerformance Analysis of Backscatter Communications with Side Information

**DOI:** 10.3390/s23208358

**Published:** 2023-10-10

**Authors:** Masoud Kaveh, Farshad Rostami Ghadi, Riku Jäntti, Zheng Yan

**Affiliations:** 1Department of Information and Communication Engineering, Aalto University, 02150 Espoo, Finland; masoud.kaveh@aalto.fi; 2Telecommunication Research Institute (TELMA), Universidad de Málaga, 29010 Málaga, Spain; farshad@ic.uma.es; 3School of Cyber Engineering, Xidian University, Xi’an 710071, China; zyan@xidian.edu.cn

**Keywords:** side information, backscatter communication, average secrecy capacity, secrecy outage probability

## Abstract

Backscatter communication (BC) systems are a promising technology for internet of things (IoT) applications that allow devices to transmit information by modulating ambient radio signals without the need for a dedicated power source. However, the security of BC systems is a critical concern due to the vulnerability of the wireless channel. This paper investigates the impact of side information (SI) on the secrecy performance of BC systems. SI mainly refers to the additional knowledge that is available to the communicating parties beyond transmitted data, which can be used to enhance reliability, efficiency, security, and quality of service in various communication systems. In particular, in this paper, by considering a non-causally known SI at the transmitter, we derive compact analytical expressions of average secrecy capacity (ASC) and secrecy outage probability (SOP) for the proposed system model to analyze how SI affects the secrecy performance of BC systems. Moreover, a Monte Carlo simulation validates the accuracy of our analytical results and reveals that considering such knowledge at the transmitter has constructive effects on the system performance and ensures reliable communication with higher rates than the conventional BC systems without SI, namely, lower SOP and higher ASC are achievable.

## 1. Introduction

Backscatter communication (BC) is a promising technology that enables devices to transmit information by modulating ambient radio signals, eliminating the need for a dedicated power source. This innovative approach has gained significant attention in internet of things (IoT) applications over recent years due to its low power requirements and potential for widespread deployment [1,2,3,4]. However, the lack of a tractable transmission structure in the BC system makes it susceptible to passive eavesdropping attacks, where an adversary can intercept and decode transmitted information. Moreover, the unlicensed and shared nature of the wireless spectrum in which BC operates raises concerns about interference and unauthorized access. Hence, ensuring the security of the BC system remains a critical challenge that should be considered in future wireless communication technologies such as sixth generation (6G) [5,6,7,8,9].

To address the aforesaid challenges, on the one hand, applying physical layer security (PLS) techniques in BC systems can be helpful [10,11,12,13,14,15,16,17]. The concept of PLS was first proposed by Shannon [18] and then studied by Wyner [19] for a basic wiretap channel exploiting Shannon’s notion of perfect secrecy. Indeed, the PLS aims to exploit the unique characteristics of the wireless channel to provide secure communication. Hence, by leveraging the inherent properties of the channel, such as fading and noise, the PLS can enhance the confidentiality and integrity of the transmitted information in BC systems. On the other hand, considering side information (SI) at the transmitters/receivers may play a vital role in improving the security and reliability of wireless BC systems. Generally speaking, SI mainly refers to additional knowledge available to the communicating parties beyond the transmitted data (e.g., channel state information (CSI), interference sources, noise statistics, prior data transmissions, etc.). The use of SI at the transmitter was first introduced by Shannon [20] for single-user point-to-point (P2P) communication systems, and then studied by Jafar [21] in multi-user communication systems. In this regard, due to the advantages of SI in reducing the destructive effects of interference and guaranteeing reliable communication with higher rates, Mitrpant et al. [22] analyzed the Gaussian wiretap channel with SI. Then, by exploiting dirty paper coding [23], Chen and Vink [24] studied the impact of SI on the Gaussian wiretap channel to find out how much secret information can be reliably and securely sent to the legitimate receiver without leaking information about the secret message to the eavesdropper. They showed that the SI at the transmitter provides a larger secrecy capacity and guarantees a more secure communication.

### 1.1. Related Works

In recent years, several contributions have been carried out to analyze the performance of secure BC systems [25,26,27,28,29,30,31,32,33,34,35,36,37,38] and to evaluate the impact of SI on various communication systems [39,40,41,42,43,44,45]. Saad et al. [25] studied the PLS performance of a wireless backscatter systems in terms of the secrecy outage probability (SOP). Zhang et al. [26] analyzed the SOP of a multi-tag BC system in the presence of an eavesdropper. Furthermore, Liu et al. [27] enhanced the SOP using an optimal tag selection scheme for a passive BC with multiple tags and one eavesdropper. In [28], Liu et al. proposed a tag selection scheme to enhance the average secrecy capacity (ASC) and SOP of a multi-tag self-powered BC system in the presence of an eavesdropper. By deriving an analytical expression of SOP, Muratkar et al. [29] studied the impact of the eavesdropper’s and reader’s motion on ambient BC systems’ secrecy performance when channel estimation is imperfect. In [30], Zheng et al. proposed an overlay cognitive ambient BC non-orthogonal multiple access (NOMA) system for intelligent transportation systems. They analyzed the secrecy performance of their proposed system model in the presence of an eavesdropping vehicle by deriving the SOP. In addition, Jia et al. [31] studied the secure multi-antenna transmission in an ambient BC-based intelligent transportation system in the presence of a passive eavesdropper with jamming, where a cooperative jammer is placed in the system to deliberately disrupt the eavesdropper without influencing the reader. For analyzing the performance of the proposed scheme, they derived a new closed-form expression of SOP.

Sharma and Kumbhani [32] conducted an in-depth analysis of secrecy performance in automatic toll collection systems employing BC. Their investigation focused on assessing the probability of secrecy compromise, incorporating the principles of PLS. The study placed particular emphasis on the effects of varying distances between the tag, reader, and eavesdropper. In [33], the emphasis shifted towards the evaluation of the secrecy performance in wireless-powered BC systems, particularly within the context of smart sustainable cities. Unique to this work was the consideration of the spatial randomness of network nodes in large-scale wireless-powered BCs. The authors employed a stochastic geometry framework to analyze SOP, accounting for imperfect successive interference cancellation and energy-harvesting constraints. A significant stride in the domain of two-way ambient BC was made by Wang et al. [34]. Their work addressed the challenge of secure communication in two-way ambient BC networks, introducing PLS considerations. The study derived analytical and asymptotic expressions for SOP, showcasing the intricate trade-offs between reliability and security. In the realm of UAV-enabled BC, Ref. [35] brought attention to the issue of information leakage in wireless channels. The authors introduced a multi-user secure BC system with analog beamforming and randomized continuous wave techniques. By exploiting randomized continuous wave techniques for eroding eavesdropping links, the study offered closed-form expressions for the secrecy rate, demonstrating substantial improvements in secrecy performance.

That et al. [36] tackled the secrecy capacity of bi-static BC networks, a critical concern in the adoption of battery-free IoT sensors. This study centered on the presence of a malicious eavesdropper and presented closed-form expressions for ASC. The results were validated through rigorous Monte Carlo simulations. The reliability and security of ambient backscatter NOMA systems under in-phase and quadrature-phase imbalance were the subject of investigation in [37]. The authors derived analytical expressions for SOP, uncovering intriguing insights into the impact of in-phase and quadrature-phase imbalance on system reliability and security trade-offs. In [38], an analysis of PLS was extended to ambient BC systems with source and reader mobility. The study employed SOP as a performance metric, exploring the effects of varying relative speed and the number of tags on the security of the system.

By considering perfect SI, Kim and Skoglund [39] maximized the expected rate over a single-input single-output (SISO) slowly fading Gaussian channel. In contrast, by assuming partial SI, Narula et al. [40] analyzed the expected signal-to-noise ratio (SNR) for a multiple antenna data transmission, where they showed that even a small amount of SI can be quite valuable for the considered system model. In addition, by considering the partial SI at the transmitter to exploit multi-user diversity, the capacity of the multiple-input and multiple-output (MIMO) broadcast channels was derived in [41]. Compact analytical expressions of the coverage region and outage probability for multiple access communications in the presence of non-causally known SI at transmitters were derived in [42,43], respectively. In addition, the impact of SI on emerging RIS-aided multiple access communications in terms of the capacity region and outage probability was investigated in [44]. Furthermore, Ghadi et al. [45] obtained closed-form expressions for the ASC and SOP under correlated Rayleigh fading channels.

While the potential of SI in providing secure and reliable communications for the next generation of wireless networks has been widely acknowledged, the specific challenges and opportunities it presents within the context of BC remain largely untapped. Addressing this research gap is imperative for advancing our understanding of SI’s impact on BC security and for shaping the future of wireless communication systems.

### 1.2. Motivations and Contributions

To the best of the authors’ knowledge, the performance analysis of secure BC systems in the presence of SI remains an open challenge. Hence, motivated by the potential of SI in providing secure and reliable communications for the next generation of wireless networks and the unique advantages of BC systems, in this paper, we investigate the efficiency of the secrecy performance of wireless BC systems when SI is available at the transmitter.

In particular, the main contributions of this paper can be summarized as follows.

By considering the non-causally known SI at the transmitter, we first derive the closed-form expressions of marginal distributions of the equivalent SNR at both a legitimate receiver and an eavesdropper under independent Rayleigh fading channels.Based on the derived probability density function (PDF) and cumulative distribution function (CDF) of the received SNR at the receiver and eavesdropper, we obtain the analytical expressions of the ASC and SOP to analyze the secrecy performance of the BC systems under the effect of non-causally known SI.Finally, we validate our analytical results through a Monte Carlo simulation. Our numerical results validate the analytical expressions and indicate that even considering a small amount of non-causally known SI at the transmitter can significantly enhance the performance of secure BC systems, namely, providing a higher ASC and a lower SOP compared with the *blank* BC scenarios (i.e., without SI).

### 1.3. Paper Organization

The rest of this paper is organized as follows. Section 2 presents the system model. Section 3 analyses the secrecy performance of the proposed system model in terms of the ASC and SOP. Section 4 provides analytical and simulation results, and Section 5 concludes the paper.

## 2. System Model

This section presents the system model, including the channel model and SNR distribution at the legitimate receiver and the eavesdropper by considering the effect of SI.

### 2.1. Channel Model

We consider a secure backscatter communication system with non-causally known SI at the legitimate transmitter (Alice), as shown in Figure 1 (It is assumed that the interfering sequence SI at Alice is injected from an external dominant source that exhibits a strong line-of-sight condition with reduced fading fluctuation (Figure 5 in [46]). Hence, we consider this unfaded counterpart approximation for the fading coefficients corresponding to the interfering signals in our analysis [45]). Alice is wirelessly powered up by a remote radio-frequency (RF) source and aims to send a confidential message to a legitimate receiver (Bob) over a wireless fading channel, while an eavesdropper (Eve) attempts to decode the message from its received signal. For simplicity, and without loss of generality, since the RF source is sending an unmodulated carrier, both Bob and Eve can employ cancellation techniques to mitigate the impact of direct link interference from the RF source [47,48,49]. Moreover, we suppose that all nodes are equipped with a single antenna, and thus, the received signal at Bob and Eve can be expressed as
(1)$$\begin{array}{c} y_{K}=h_{T}h_{K}x+s+z_{K},\;K\in \left\{B,E\right\} \end{array}$$
where $h_{T}$ denotes the fading channel coefficient between the PB and Alice, $h_{K}$ is the fading channel coefficient between Alice and the node *K*, *s* defines the non-causally known SI at Alice with variance *Q* (i.e., $s∼N\left(0,Q\right)$) which is independently and identically distributed (i.i.d) with probability distribution p(s), and $z_{K}$ corresponds to i.i.d. additive white Gaussian noise (AWGN) with zero mean and variance $\sigma_{K}^{2}$ at the node *K*. Furthermore, the instantaneous received signal power at Alice is given by
(2)$$\begin{array}{c} P_{A}=P_{T}L_{T}d_{T}^{-\alpha }{\left|h_{T}\right|}^{2}, \end{array}$$
in which $P_{T}$ is the transmitted power by the PB, $L_{T}$ incorporates the gains of the transmit and receive antennas and frequency-dependent propagation losses, $d_{T}$ is the distance between the PB and Alice, and α>2 is the path-loss exponent. Hence, the instantaneous SNR at node *K* can be defined as
(3)$$\begin{array}{c} \gamma_{K}=\frac{P_{A}{\left|h_{K}\right|}^{2}}{\sigma_{K}^{2}}=\frac{P_{T}L_{T}}{\sigma_{K}^{2}d_{T}^{\alpha }d_{K}^{\alpha }}|h_{T}{|}^{2}{\left|h_{K}\right|}^{2}={\bar{\gamma }}_{K}W_{K}, \end{array}$$
in which ${\bar{\gamma }}_{K}=\frac{P_{T}L_{T}}{\sigma_{K}^{2}d_{T}^{\alpha }d_{K}^{\alpha }}$ is the average SNR.

### 2.2. SNR Distribution

In this paper, we assume that all fading channels follow a Rayleigh distribution, therefore, the channel power gains $g_{v}={\left|h_{v}\right|}^{2}$, for v∈{T,K}, are the exponential distribution with unit mean (i.e., $E[g_{v}]=1$). Since the SNR $\gamma_{K}$ includes the product of two independent exponential random variables, the PDF of $W_{K}=g_{T}g_{K}$ can be mathematically determined as
(4)$$\begin{array}{cc} \;\;\;\;\;\;f_{W_{K}}\left(w_{K}\right) & =\int_{0}^{\infty }\frac{1}{g_{T}}f_{G_{T}}\left(g_{T}\right)f_{G_{K}}\left(w_{K}/g_{T}\right)dg_{T} \end{array}$$
(5)$$\begin{array}{cc} \; & =\int_{0}^{\infty }\frac{1}{g_{T}}e^{-g_{T}-\frac{w_{K}}{g_{T}}}dg_{T} \end{array}$$
(6)$$\begin{array}{cc} \; & \;=2K_{0}\left(2\sqrt{w_{K}}\right), \end{array}$$
where $K_{\nu }\left(.\right)$ is the modified Bessel function of the second kind. Then, exploiting the fact that $f_{\gamma_{K}}\left(\gamma_{K}\right)=\frac{1}{{\bar{\gamma }}_{K}}f_{W_{K}}\left(\frac{\gamma_{K}}{{\bar{\gamma }}_{K}}\right)$, the PDF of $\gamma_{K}$ can be given by
(7)$$\begin{array}{c} f_{\gamma_{K}}\left(\gamma_{K}\right)=\frac{2}{{\bar{\gamma }}_{K}}K_{0}\left(2\sqrt{\frac{\gamma_{K}}{{\bar{\gamma }}_{K}}}\right). \end{array}$$
By using the definition, $F_{X}\left(x\right)=\int_{0}^{x}f_{X}\left(x\right)dx$, the CDF of the SNR $\gamma_{K}$ can be derived as
(8)$$\begin{array}{c} F_{\gamma_{K}}\left(\gamma_{K}\right)=1-2\sqrt{\frac{\gamma_{K}}{{\bar{\gamma }}_{K}}}K_{1}\left(2\sqrt{\frac{\gamma_{K}}{{\bar{\gamma }}_{K}}}\right). \end{array}$$

## 3. Secrecy Performance Analysis

Here, we derive the compact analytical expressions of the ASC and SOP for the proposed system model under Rayleigh fading channels.

### 3.1. ASC Analysis

In order to analyze the ASC, we first need to define the secrecy capacity for the considered system model. The secrecy capacity for a classic wiretap channel from the information theory viewpoint was derived in [24] and then extended to the wireless channels by considering the propagation environment effects such as fading, shadowing, path-loss, etc. [45]. Hence, the secrecy capacity for the block fading wiretap channel with non-causally known SI at the transmitter is given by (Theorem 2 in [45])
(9)$$\begin{array}{c} C_{s}=\left\{\begin{array}{cc} \log_{2}\left(1+\gamma_{B}\right) & \mathrm{if}\;\mathrm{Corollary}\;1\;\\ \log_{2}\left(\frac{1+{\bar{\gamma }}_{s,B}+\gamma_{B}}{1+{\bar{\gamma }}_{s,E}+\gamma_{E}}\right) & \mathrm{if}\;\mathrm{Corollary}\;2 \end{array}\right., \end{array}$$
where ${\bar{\gamma }}_{s,K}=\frac{Q}{\sigma_{K}^{2}}$. blueBesides, Corollaries 1 and 2 are defined in [45].

**Theorem 1**.
*The ASC for the concerned backscatter communications with non-causally known SI at the transmitter is given by*

(10)
$$\begin{array}{c} {\bar{C}}_{s}=\left\{\begin{array}{cc} {\bar{C}}_{s}^{1} & \mathrm{if}\;\mathrm{Corollary}\;1\\ {\bar{C}}_{s}^{2} & \mathrm{if}\;\mathrm{Corollary}\;2 \end{array}\right., \end{array}$$


*where*

(11)
$$\begin{array}{c} {\bar{C}}_{s}^{1}=\frac{1}{{\bar{\gamma }}_{B}\ln2}G_{1,3}^{3,1}\left(\frac{1}{{\bar{\gamma }}_{B}}|\begin{array}{c} -1\\ -1,-1,0 \end{array}\right), \end{array}$$


*and*

(12)
       C¯s2=C1∑n=1∞∑l=0n(−1)nnnlγ¯Blγ¯s,Bn−lG2,22,2γ¯Bγ¯E|−12−l,−12−l12,−12


(13)
       +C2∑n=1∞∑l=0n(−1)nnnlγ¯Elγ¯s,En−lG2,22,2γ¯Eγ¯B|−12−l,−12−l12,−12


(14)
$$\begin{array}{cc} & \;-C_{3}\;H_{1,0:1,2;0,1}^{0,1:2,1;1,0}\left(\;\begin{array}{c} \;-\frac{1}{{\bar{\gamma }}_{s,B}}\\ -\frac{{\bar{\gamma }}_{B}}{{\bar{\gamma }}_{s,B}} \end{array}\;|\;\begin{array}{c} (0;1,1):(0,1);--\;\\ --:(0,1),(0,1);(0,1) \end{array}\;\right), \end{array}$$

*in which $C_{1}=\frac{\sqrt{{\bar{\gamma }}_{B}}}{{\bar{\gamma }}_{E}\ln2}$, $C_{2}=\frac{\sqrt{{\bar{\gamma }}_{E}}}{{\bar{\gamma }}_{B}\ln2}$, and $C_{3}=\frac{{\bar{\gamma }}_{s,B}}{{\bar{\gamma }}_{B}\ln2}$.*


**Proof**.By re-expressing the logarithm function in terms of the Meijer’s G-function, inserting the PDF of ${\bar{\gamma }}_{B}$ from (Equation 7), and using the integral format provided in [50], Equation (2.24.4.3), ${\bar{C}}_{s}^{1}$ can be derived as follows:
(15)$$\begin{array}{cc} \;\;{\bar{C}}_{s}^{1} & =\int_{0}^{\infty }\log_{2}\left(1+\gamma_{B}\right)f_{B}\left(\gamma_{B}\right)d\gamma_{B} \end{array}$$
(16)$$\begin{array}{cc} \; & =\frac{2}{{\bar{\gamma }}_{B}\ln2}\int_{0}^{\infty }G_{2,2}^{1,2}\left(\gamma_{B}|\begin{array}{c} 1,1\\ 1,0 \end{array}\right)K_{0}\left(2\sqrt{\frac{\gamma_{B}}{{\bar{\gamma }}_{B}}}\right)d\gamma_{B} \end{array}$$
(17)$$\begin{array}{cc} \; & \;\;\;=\frac{1}{{\bar{\gamma }}_{B}\ln2}G_{1,3}^{3,1}\left(\frac{1}{{\bar{\gamma }}_{B}}|\begin{array}{c} -1\\ -1,-1,0 \end{array}\right). \end{array}$$
In order to prove the derived analytical expression for ${\bar{C}}_{s}^{2}$ under the SI effect we start with the ASC formula [45].
(18)$$\begin{array}{cc} \;\;\;\;\;\;\;{\bar{C}}_{s}^{2} & =\int_{0}^{\infty }\int_{0}^{\infty }\log_{2}\left(\frac{1+{\bar{\gamma }}_{s,B}+\gamma_{B}}{1+{\bar{\gamma }}_{s,E}+\gamma_{E}}\right)f_{B}\left(\gamma_{B}\right)f_{E}\left(\gamma_{E}\right)d\gamma_{B}d\gamma_{E}\\ \; & =\int_{0}^{\infty }\log_{2}\left(1+{\bar{\gamma }}_{s,B}+\gamma_{B}\right)f_{\gamma_{B}}\left(\gamma_{B}\right)F_{\gamma_{E}}\left(\gamma_{B}\right)d\gamma_{B}\\ \; & +\int_{0}^{\infty }\log_{2}\left(1+{\bar{\gamma }}_{s,E}+\gamma_{E}\right)f_{\gamma_{E}}\left(\gamma_{E}\right)F_{\gamma_{B}}\left(\gamma_{E}\right)d\gamma_{E} \end{array}$$
(19)$$\begin{array}{cc} \; & \;\;-\int_{0}^{\infty }\log_{2}\left(1+{\bar{\gamma }}_{s,E}+\gamma_{E}\right)f_{\gamma_{E}}\left(\gamma_{E}\right)d\gamma_{E}. \end{array}$$By inserting (Equation 7) and (Equation 8) into (Equation 19), we will need to compute ${\bar{C}}_{s}^{2}$ as
(20)$$\begin{array}{c} {\bar{C}}_{s}^{2}=I_{1}-I_{2}-I_{3}, \end{array}$$
where
(21)$$\begin{array}{c} I_{1}=\frac{2}{{\bar{\gamma }}_{B}}\int_{0}^{\infty }\log_{2}\left(1+{\bar{\gamma }}_{s,B}+\gamma_{B}\right)K_{0}\left(2\sqrt{\frac{\gamma_{B}}{{\bar{\gamma }}_{B}}}\right)d\gamma_{B}, \end{array}$$
(22)$$\begin{array}{c} I_{2}=\frac{4}{{\bar{\gamma }}_{E}}\int_{0}^{\infty }\log_{2}\left(1+{\bar{\gamma }}_{s,B}+\gamma_{B}\right)\sqrt{\gamma_{B}}\;K_{0}\left(2\sqrt{\frac{\gamma_{B}}{{\bar{\gamma }}_{B}}}\right)K_{1}\left(2\sqrt{\frac{\gamma_{B}}{{\bar{\gamma }}_{E}}}\right)d\gamma_{B}, \end{array}$$
(23)$$\begin{array}{c} I_{3}=\frac{4}{{\bar{\gamma }}_{B}}\int_{0}^{\infty }\log_{2}\left(1+{\bar{\gamma }}_{s,E}+\gamma_{E}\right)\sqrt{\gamma_{E}}\;K_{0}\left(2\sqrt{\frac{\gamma_{E}}{{\bar{\gamma }}_{E}}}\right)K_{1}\left(2\sqrt{\frac{\gamma_{E}}{{\bar{\gamma }}_{B}}}\right)d\gamma_{E}. \end{array}$$In order to solve $I_{1}$, we first use the Meijer’s G-function shape of the logarithm and MacDonald’s functions (see (Equation 24) and (Equation 25)) and the $y={\bar{\gamma }}_{s,B}+\gamma_{B}$ variable change.
(24)$$\begin{array}{c} \ln\left(1+x\right)=G_{1,2}^{2,2}\left(x|\begin{array}{c} 1,1\\ 1,0 \end{array}\right). \end{array}$$
(25)$$\begin{array}{c} K_{\nu }\left(x\right)=\frac{1}{2}G_{0,2}^{2,0}\left(\frac{x^{2}}{4}|\begin{array}{c} -\\ \frac{\nu }{2},-\frac{\nu }{2} \end{array}\right). \end{array}$$Then, we have $I_{1}$ as
(26)$$\begin{array}{c} I_{1}=\frac{1}{{\bar{\gamma }}_{B}\ln2}\int_{0}^{\infty }G_{1,2}^{2,2}\left(y|\begin{array}{c} 1,1\\ 1,0 \end{array}\right)G_{0,2}^{2,0}\left(\frac{y-{\bar{\gamma }}_{s,B}}{{\bar{\gamma }}_{B}}|\begin{array}{c} -\\ 0,0 \end{array}\right)dy. \end{array}$$By using the the Meijer’s G-function definition, we have (Equation 26) as
(27)$$\begin{array}{c} I_{1}=\frac{1}{{\bar{\gamma }}_{B}\ln2{\left(2\pi j\right)}^{2}}\underset{I_{1}^{\prime }}{\underset{︸}{\int_{0}^{\infty }y^{\zeta_{1}}{\left(\frac{y}{{\bar{\gamma }}_{B}}-\frac{{\bar{\gamma }}_{s,B}}{{\bar{\gamma }}_{B}}\right)}^{\zeta_{2}}dy}}\oint_{L_{1}}\oint_{L_{2}}\frac{\Gamma \left(1-\zeta_{1}\right)\Gamma^{2}\left(\zeta_{1}\right)}{\Gamma \left(1+\zeta_{1}\right)}\Gamma^{2}\left(-\zeta_{2}\right)d\zeta_{2}d\zeta_{1}, \end{array}$$
where $L_{1}$ and $L_{2}$ are special contours and $I_{1}^{\prime }$ can be obtained as
(28)$$\begin{array}{c} I_{1}^{\prime }={\left(-{\bar{\gamma }}_{s,B}\right)}^{1+\zeta_{1}}{\left(-\frac{{\bar{\gamma }}_{s,B}}{{\bar{\gamma }}_{B}}\right)}^{\zeta_{2}}\;\frac{\Gamma \left(1-\zeta_{1}-\zeta_{2}\right)\Gamma \left(1+\zeta_{1}\right)}{\Gamma \left(-\zeta_{2}\right)}. \end{array}$$By putting (Equation 28) into (Equation 27) and changing $\zeta_{1}=-\zeta_{1}$ and $\zeta_{2}=-\zeta_{2}$, we have
(29)$$\begin{array}{c} I_{1}=\frac{-{\bar{\gamma }}_{s,B}}{{\bar{\gamma }}_{B}\ln2{\left(2\pi j\right)}^{2}}\oint_{L_{1}}\oint_{L_{2}}\Gamma \left(1+\zeta_{1}+\zeta_{2}\right)\Gamma \left(1+\zeta_{1}\right)\Gamma^{2}\left(-\zeta_{1}\right){\left(-\frac{1}{{\bar{\gamma }}_{s,B}}\right)}^{\zeta_{1}}\Gamma \left(-\zeta_{2}\right){\left(-\frac{{\bar{\gamma }}_{B}}{{\bar{\gamma }}_{s,B}}\right)}^{\zeta_{2}}d\zeta_{2}d\zeta_{1}. \end{array}$$According to the definition of the bivariate Fox’s H-function ([51], Equations (2.56)–(2.60)), we can write (Equation 29) as (Equation 14), so the proof is completed for $I_{1}$.In order to compute $I_{2}$, we first use the Taylor distribution of the logarithm and binomial functions and also re-express the MacDonald’s functions in terms of the Meijer’s G-function. Therefore, we can rewrite (Equation 22) as
(30)I2=I∑n=1∞∑l=0n(−1)n+1nnlγ¯s,Bn−l∫0∞γB12+lG2,00,2γBγB¯|−0,0G0,22,0γBγ¯E|−0,0dγB︸I2′,
where $I=\frac{1}{{\bar{\gamma }}_{B}\sqrt{{\bar{\gamma }}_{E}}\ln2}$. By using the ([50], Equation (2.24.3.1)), we can obtain $I_{2}^{\prime }$ as
(31)$$\begin{array}{c} I_{2}^{\prime }={\left({\bar{\gamma }}_{B}\right)}^{\frac{3}{2}+l}\;G_{2,2}^{2,2}\left(\;\begin{array}{c} \frac{{\bar{\gamma }}_{B}}{{\bar{\gamma }}_{E}} \end{array}\;|\;\begin{array}{c} -\frac{1}{2}-l,-\frac{1}{2}-l\;\\ \frac{1}{2},-\frac{1}{2} \end{array}\;\right). \end{array}$$Now, by inserting (Equation 31) into (Equation 30), we will have $I_{2}$ as (Equation 12) and the proof is completed. We can also take a similar step for completing the proof for obtaining $I_{3}$ as (Equation 13). As a result, the proof will be completed for ${\bar{C}}_{s}^{2}$.□

### 3.2. SOP Analysis

The SOP is defined as the probability that the random secrecy capacity $C_{s}$ is less than a target secrecy rate $R_{s}>0$, i.e.,
(32)$$\begin{array}{c} P_{sop}=\mathrm{Pr}\left(C_{s}\leq R_{s}\right). \end{array}$$

**Theorem 2**.
*The SOP for the considered backscatter communications with non-causally known SI at the transmitter is given by*

(33)
$$\begin{array}{c} P_{sop}=\left\{\begin{array}{cc} P_{sop}^{1} & \mathrm{if}\;\mathrm{Corollary}\;1\\ P_{sop}^{2} & \mathrm{if}\;\mathrm{Corollary}\;2 \end{array}\right., \end{array}$$

*where*

(34)
$$\begin{array}{c} P_{sop}^{1}=1-2\sqrt{\frac{R_{th}}{{\bar{\gamma }}_{B}}}K_{1}\left(2\sqrt{\frac{R_{th}}{{\bar{\gamma }}_{B}}}\right), \end{array}$$

*and*

(35)
$$\begin{array}{c} P_{sop}^{2}=1-\frac{R_{t}^{\prime \frac{3}{2}}}{R_{t}{\bar{\gamma }}_{E}}\;H_{1,0:1,0;2,1}^{0,1:0,1;1,2}\left(\;\begin{array}{c} \;\frac{{\bar{\gamma }}_{B}}{R_{t}^{\prime }}\\ \frac{R_{t}{\bar{\gamma }}_{E}}{R_{t}^{\prime }} \end{array}\;|\;\begin{array}{c} \left(2;1,1\right):\left(\frac{1}{2},1\right);\left(1,1\right),\left(1,1\right)\;\\ --:--;\left(1,1\right) \end{array}\;\right). \end{array}$$



**Proof**.By substituting $C_{s}^{1}$ into (Equation 32), $P_{sop}^{1}$ can be derived as
(36)$$\begin{array}{cc} \;\;\;\;\;\;P_{sop}^{1} & =\mathrm{Pr}\left(\log_{2}\left(1+\gamma_{B}\right)\leq R_{s}\right) \end{array}$$
(37)$$\begin{array}{cc} \; & =\mathrm{Pr}\left(\gamma_{B}\leq 2^{R_{s}}-1\right) \end{array}$$
(38)$$\begin{array}{cc} \; & =F_{\gamma_{B}}\left(R_{th}\right), \end{array}$$
in which by inserting $R_{th}=2^{R_{s}}-1$ into (Equation 8), the proof is completed.Similarly, by inserting $C_{s}^{2}$ into the SOP definition and considering the marginal distributions, we have
(39)$$\begin{array}{cc} \;\;\;\;\;P_{sop}^{2} & =\mathrm{Pr}\left(\log_{2}\left(\frac{1+{\bar{\gamma }}_{s,B}+\gamma_{B}}{1+{\bar{\gamma }}_{s,E}+\gamma_{E}}\right)>R_{s}\right) \end{array}$$
(40)$$\begin{array}{cc} \; & =\mathrm{Pr}\left(\gamma_{B}>2^{R_{s}}\left(1+{\bar{\gamma }}_{s,E}+\gamma_{E}\right)-\left(1+{\bar{\gamma }}_{s,B}\right)\right) \end{array}$$
(41)$$\begin{array}{cc} \; & \;=\int_{0}^{\infty }F_{B}\left(\gamma_{th}\right)f_{E}\left(\gamma_{E}\right)d\gamma_{E} \end{array}$$
(42)$$\begin{array}{cc} \; & \;=1-\frac{4}{{\bar{\gamma }}_{E}}\int_{0}^{\infty }\sqrt{\frac{\gamma_{th}}{{\bar{\gamma }}_{B}}}K_{1}\left(2\sqrt{\frac{\gamma_{th}}{{\bar{\gamma }}_{B}}}\right)K_{0}\left(2\sqrt{\frac{\gamma_{E}}{{\bar{\gamma }}_{E}}}\right)d\gamma_{E}, \end{array}$$
where $\gamma_{th}=R_{t}\gamma_{E}+R_{t}^{\prime }$, $R_{t}=2^{R_{s}}$, and $R_{t}^{\prime }=2^{R_{s}}\left(1+{\bar{\gamma }}_{s,E}\right)-1-{\bar{\gamma }}_{s,B}$. Now, by re-expressing the Bessel functions in terms of the Meijer’s G-function we have
(43)$$\begin{array}{cc} P_{sop}^{2} & =1-\frac{1}{{\bar{\gamma }}_{E}}\int_{0}^{\infty }\sqrt{\frac{R_{t}\gamma_{E}+R_{t}^{\prime }}{{\bar{\gamma }}_{B}}}\\ \; & \times \;G_{0,2}^{2,0}\left(\frac{R_{t}\gamma_{E}+R_{t}^{\prime }}{{\bar{\gamma }}_{B}}|\begin{array}{c} -\\ \frac{1}{2},-\frac{1}{2} \end{array}\right)G_{0,2}^{2,0}\left(\frac{\gamma_{E}}{{\bar{\gamma }}_{E}}|\begin{array}{c} -\\ 0,0 \end{array}\right)d\gamma_{E}. \end{array}$$By using the Meijer G-function definition, we can rewrite (Equation 43) as
(44)$$\begin{array}{cc} P_{sop}^{2} & =1-\frac{1}{{\bar{\gamma }}_{E}{\left(2\pi j\right)}^{2}}\underset{J_{1}}{\underset{︸}{\int_{0}^{\infty }{\left(\sqrt{\frac{R_{t}\gamma_{E}+R_{t}^{\prime }}{{\bar{\gamma }}_{B}}}\right)}^{\frac{1}{2}+\zeta_{1}}{\left(\frac{\gamma_{E}}{{\bar{\gamma }}_{E}}\right)}^{\zeta_{2}}d\gamma_{E}}}\\ & \times \oint_{L_{1}}\oint_{L_{2}}\Gamma \left(\frac{1}{2}-\zeta_{1}\right)\Gamma \left(-\frac{1}{2}-\zeta_{1}\right)\Gamma^{2}\left(-\zeta_{2}\right)d\zeta_{2}d\zeta_{1}, \end{array}$$
where we can obtain $J_{1}$ as
(45)$$\begin{array}{c} J_{1}=\frac{{R^{\prime }}_{t}^{\frac{3}{2}}}{R_{t}}{\left(\frac{R_{t}^{\prime }}{{\bar{\gamma }}_{B}}\right)}^{\zeta_{1}}{\left(\frac{R_{r}^{\prime }ime_{t}}{R_{t}{\bar{\gamma }}_{E}}\right)}^{\zeta_{2}}\;\frac{\Gamma \left(-3-\zeta_{1}-\zeta_{2}\right)\Gamma \left(1+\zeta_{2}\right)}{\Gamma \left(-\frac{1}{2}-\zeta_{1}\right)}. \end{array}$$Now, by inserting (Equation 45) into (Equation 44) and changing $\zeta_{1}=-\zeta_{1}$ and $\zeta_{2}=-\zeta_{2}$, we have $P_{sop}^{2}$ as
(46)$$\begin{array}{cc} P_{sop}^{2} & =1-P\oint_{L_{1}}\oint_{L_{2}}\Gamma \left(-3+\zeta_{1}+\zeta_{2}\right)\Gamma \left(\frac{1}{2}+\zeta_{1}\right){\left(\frac{R_{t}^{\prime }}{{\bar{\gamma }}_{B}}\right)}^{\zeta_{1}}\Gamma \left(1+\zeta_{2}\right)\Gamma^{2}\left(-\zeta_{2}\right){\left(\frac{R_{t}^{\prime }}{R_{t}{\bar{\gamma }}_{E}}\right)}^{\zeta_{2}}d\zeta_{2}d\zeta_{1}. \end{array}$$
where $P=\frac{{R^{\prime }}_{t}^{\frac{3}{2}}}{R_{t}{\bar{\gamma }}_{E}{\left(2\pi j\right)}^{2}}$. According to the definition of the bivariate Fox’s H-function ([51], Equations (2.56)–(2.60)), we can write (Equation 46) as (Equation 35), so the proof is completed for $P_{sop}^{2}$.□

## 4. Numerical Results

In this section, we evaluate the accuracy of our analytical results with Monte Carlo simulations in the presence/absence of non-causally known SI at the transmitter. To this end, we set $P_{T}=20,\;30,\;40,\;50$ dBm, $\sigma_{B}^{2}=-50$ dBm, $\sigma_{E}^{2}=-40$ dBm, $d_{T}=d_{B}=d_{E}=10$ m, Q=−40 dBm, $R_{s}=1$ bits, α=3.

Figure 2 shows the behavior of the ASC in terms of ${\bar{\gamma }}_{B}$ for given values of ${\bar{\gamma }}_{E}$ when the non-causally known SI is available at Alice. As anticipated, it becomes evident that as ${\bar{\gamma }}_{B}$ grows, the ASC also increases. This phenomenon can be attributed to the improvement in the primary communication link (from Alice to Bob) conditions. In simpler terms, as the average SNR at Bob improves, the ability to transmit secure information between Alice and Bob becomes more effective, resulting in an increase in the ASC. It can be also observed that the presence of SI has a constructive impact on the ASC performance. In other words, when SI is available, the ASC is significantly higher compared to scenarios where SI does not exist. This implies that SI plays a vital role in enhancing the ASC, making the communication link more secure and capable of transmitting information more effectively. These observations are in alignment with the findings of [24,45], where it was shown that the non-causally known SI at the transmitter can help to achieve a larger secrecy capacity and achieve a larger rate equivocation region. Additionally, it is worth noting that the positive influence of SI on the ASC gradually diminishes as ${\bar{\gamma }}_{B}$ increases. In simpler terms, when the primary link’s signal quality from Alice to Bob is already quite strong (high ${\bar{\gamma }}_{B}$), the additional benefit provided by SI becomes less significant. This suggests that while SI remains beneficial in lower SNR scenarios, its impact becomes less pronounced as the communication conditions improve. However, it is seen that the SI does not affect the ASC performance if Corollary 1 in [45] is considered since ${\bar{C}}_{s}^{1}$ is independent of SI and only depends on $\gamma_{B}$ (see (Equation 9)). In order to gain more insights into the impact of SI on the ASC performance, Figure 3 indicates the efficiency of the ASC versus the SI ratio ${\bar{\gamma }}_{s,B}/{\bar{\gamma }}_{s,E}$ for three different scenarios ${\bar{\gamma }}_{B}>{\bar{\gamma }}_{E}$, ${\bar{\gamma }}_{B}={\bar{\gamma }}_{E}$, and ${\bar{\gamma }}_{B}<{\bar{\gamma }}_{E}$. It is obvious that in all three scenarios, ${\bar{C}}_{s}^{1}$ is fixed as the SI ratio changes and only depends on ${\bar{\gamma }}_{B}$. In contrast, it can be seen that ${\bar{C}}_{s}^{2}$ monotonically increases as the SI ratio grows for all scenarios. It is worth noting that even under the scenario that the eavesdropper link (Alice-to-eavesdropper) is better than the main link (i.e., ${\bar{\gamma }}_{B}<{\bar{\gamma }}_{E}$), the ASC is still achievable, though with lower values.

The impact of SI on the SOP performance in terms of ${\bar{\gamma }}_{B}$ for selected values of ${\bar{\gamma }}_{E}$ is illustrated in Figure 4. Due to the definition of the secrecy capacity from (Equation 9), we can see that the SOP is constant as ${\bar{\gamma }}_{B}$ grows under Col. 1 provided in [45]; however, it is observed that the SI can significantly improve the SOP performance if Col. 2 in [45] is considered as it can reduce the likelihood of communication failures in the presence of eavesdroppers. To further evaluate the impact of SI on the SOP performance, Figure 5 demonstrates the SOP behavior versus the SI ratio ${\bar{\gamma }}_{s,B}/{\bar{\gamma }}_{s,E}$ for three different scenarios. For all three scenarios, it can be observed that the SOP decreases as the SI ratio increases. We can also see that when the main link is better than the eavesdropper link (i.e., ${\bar{\gamma }}_{B}>{\bar{\gamma }}_{E}$), the lowest SOP is achievable compared with two other cases. Hence, we can see that considering the non-causally known SI at the transmitter in the secure backscatter communication system can remarkably enhance the SOP performance.

## 5. Conclusions

In this study, we have explored the impact of side information on the secrecy performance of backscatter communication systems, assuming Rayleigh fading distributions for all channel conditions. Specifically, our investigation focused on scenarios where non-causally known side information is available at the transmitter. We conducted a comprehensive analysis of the system’s secrecy performance, providing concise analytical expressions for both ASC and SOP. Our analytical findings, coupled with simulation results, have consistently demonstrated the tangible benefits of incorporating SI in securing backscatter communication systems. This inclusion of SI has proven to be highly advantageous, resulting in a significant enhancement in the system’s secrecy performance, as quantified by improvements in both ASC and SOP. To put it succinctly, our results underscore the constructive influence of non-causally known SI in enhancing the security of backscatter communication systems, particularly in scenarios characterized by strong interference. This positive effect becomes particularly pronounced when contrasted with the scenario of an interference-free, clean backscatter communication system.

## Figures and Tables

**Figure 1 sensors-23-08358-f001:**
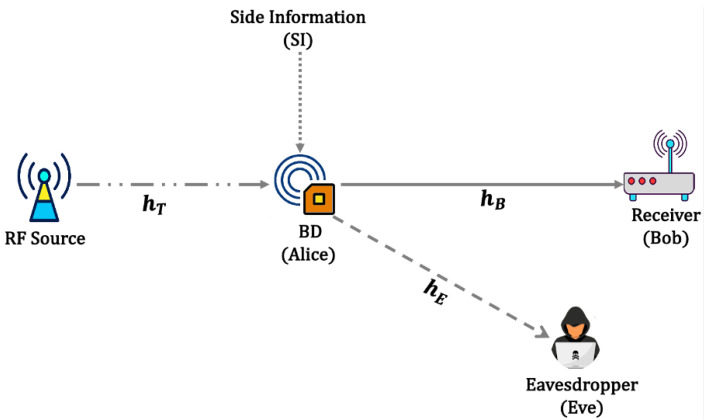
Illustration of the secure backscatter communication with SI.

**Figure 2 sensors-23-08358-f002:**
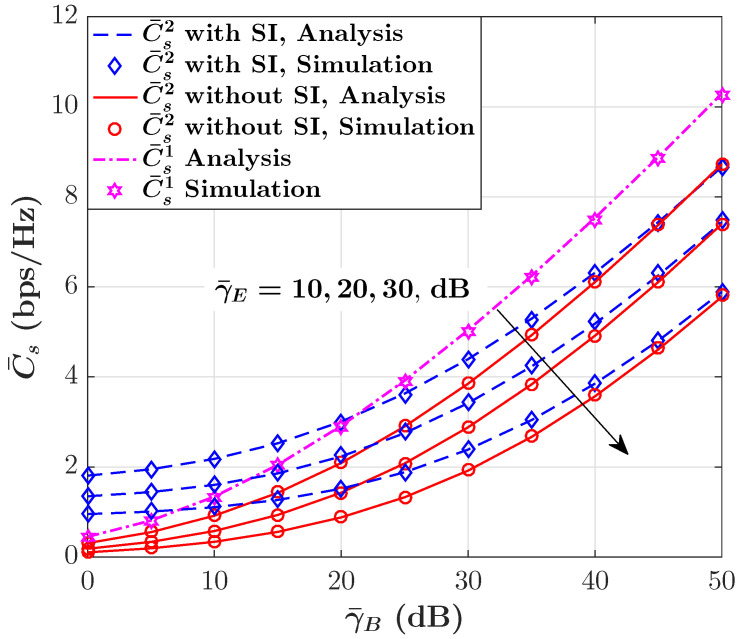
ASC versus ${\bar{\gamma }}_{B}$ for selected values of ${\bar{\gamma }}_{E}$ in presence/absence of SI.

**Figure 3 sensors-23-08358-f003:**
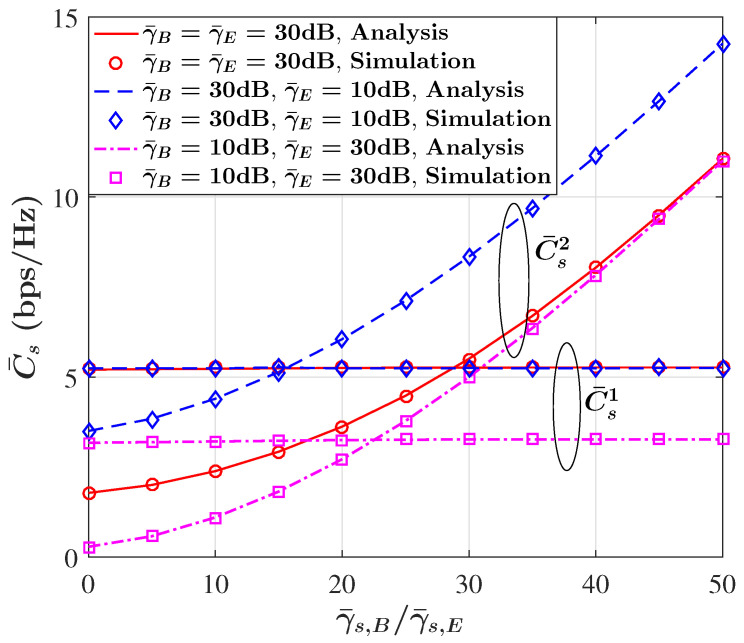
ASC versus SI ratio for three different scenarios ${\bar{\gamma }}_{B}>{\bar{\gamma }}_{E}$, ${\bar{\gamma }}_{B}={\bar{\gamma }}_{E}$, and ${\bar{\gamma }}_{B}<{\bar{\gamma }}_{E}$.

**Figure 4 sensors-23-08358-f004:**
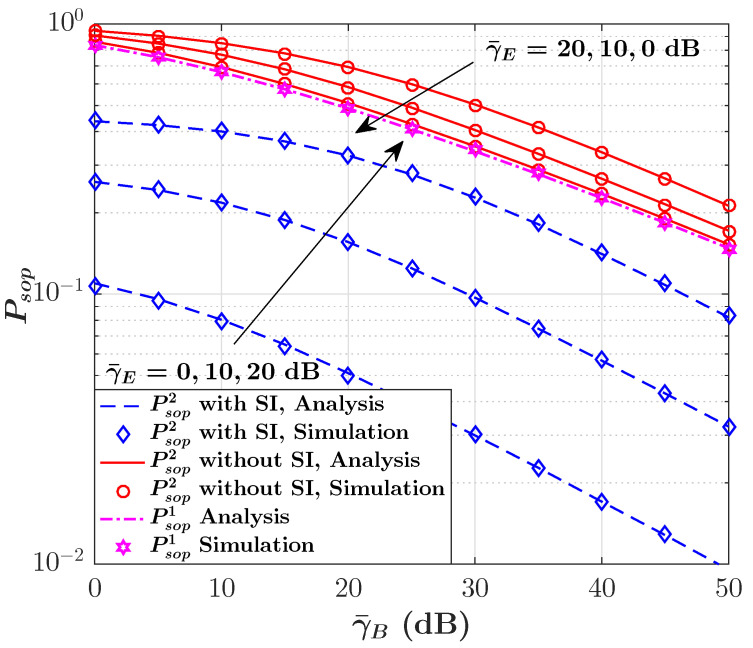
SOP versus ${\bar{\gamma }}_{B}$ for selected values of ${\bar{\gamma }}_{E}$ in presence/absence of SI.

**Figure 5 sensors-23-08358-f005:**
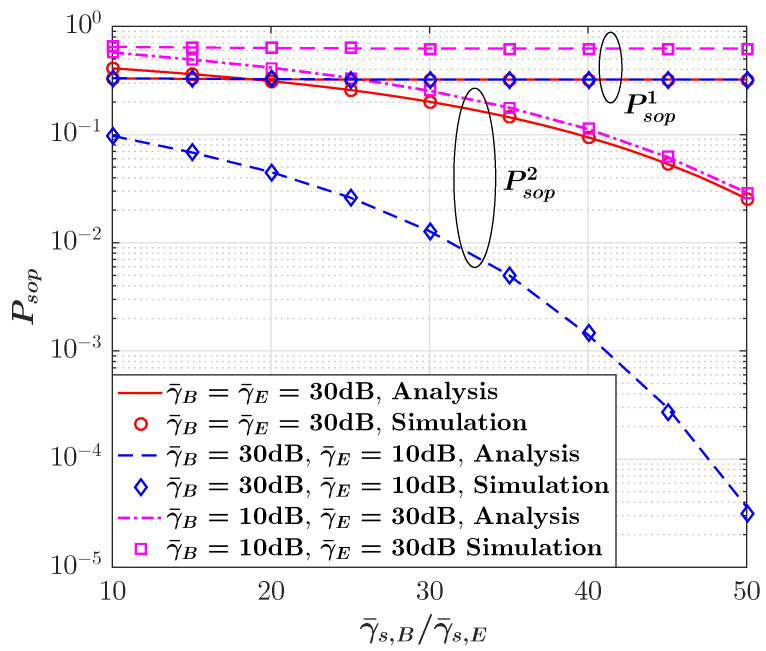
SOP versus SI ratio for three different scenarios ${\bar{\gamma }}_{B}>{\bar{\gamma }}_{E}$, ${\bar{\gamma }}_{B}={\bar{\gamma }}_{E}$, and ${\bar{\gamma }}_{B}<{\bar{\gamma }}_{E}$.

## Data Availability

Not applicable.

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
