# Peer review of "SecrecyPerformance Analysis of Backscatter Communications with Side Information"

_sensors, 2023, doi:10.3390/s23208358_

Round 1

Reviewer 1 Report

This paper is entitled “Effects of Side Information on Secure Backscatter Communication Systems” and presents the results of implementing Backscatter communications for IoT and 6G while investigating the effects of Side Information on secrecy and performance analysis. For this purpose and based on the system model proposed, they have presented and mathematically proved two theorems. Simulation results, using Monte Carlo simulation, are accurate and validate their results.

The paper is well referenced and presented, and the English writing is also very good.

For these reasons, I agree with its publication in Sensors.

Reviewer 2 Report

This paper investigates the impact of side information on the secrecy performance of backscatter systems. The derivation are solid and correct. The topic is timely and interesting. I have no comments.

The quality of English in this paper is high, and  no changes are needed. 

Reviewer 3 Report

In this paper, the authors investigate the impact of side information (SI) on the secrecy performance of BC systems. By considering a non-causally known SI at the transmitter, the authors derive compact analytical expressions of average secrecy capacity (ASC) and secrecy outage probability (SOP) for the proposed system model to analyze how SI affects the secrecy performance of BC systems. I list my main concerns below. 1. Related works are not completely reviewed. More related works should be introduced to draw a big picture for the readers. Besides, authors should compare the differences and shortcomings of different works, rather than simply listing and summarizing them. 2. There are some typos; authors need to check the paper thoroughly. 3. Extensive simulations have been carried out to validate the effectiveness of the proposed method, and many experimental results are shown in the paper. However, few result analyses are given in the paper to explain the reason why the proposed approach can outperform benchmarking methods. 4.The authors should clearly point out the major contributions of this paper by using 3 to 5 brief bullet points.  5. Some state-of-the-art references are missing in this version. The authors may check the following papers: Reliability-Driven End–End–Edge Collaboration for Energy Minimization in Large-Scale Cyber-Physical Systems," in IEEE Transactions on Reliability, doi: 10.1109/TR.2023.3297124.

Minor editing of English language required

Reviewer 4 Report

This paper investigates the impact of side information (SI) on the secrecy performance of BC systems. However, some descriptions are not clear. Some revisions are necessary in the manuscript.

1. The article has a lot of parameters and abbreviations, please make sure that all abbreviations are necessary and the parameters are defined.

2. Please elaborate further on the core innovation of the article.

3. The conclusion suggests adding more data content.

4. Reference 29 is heavily cited in this paper. Please further explain the relationship between reference 29 and this paper.

5. In the paper, authors have mentioned applying physical layer security (PLS) techniques in BC systems can be helpful. Physical layer security techniques needs to be analyzed to indicate advantages of your work, which can refer to

[a] IEEE Transactions on Industrial Informatics, vol. 18, no. 2, pp. 835-846, 2022

[b] IEEE Transactions on Industrial Informatics, 2023, DOI: 10.1109/TII.2023.3241682

[c] IEEE Transactions on Vehicular Technology, vol. 65, no. 3, pp. 1835-1841, March 2016

[d] IEEE Wireless Communications Letters, vol. 10, no. 11, pp. 2398-2401, Nov. 2021

A proof reading is needed.

Reviewer 5 Report

-        Revise the title to make it more meaningful. 

-        Explain novelty of your work presented in this work.

-        Improve the quality of figures and explain those properly.  

-        Paper needs to polish and provide a detailed explication of theoretical aspects such as conditions and theorems, and practical issues like algorithms, rules and possible applications.

-       improve the readability of the manuscript in terms of typos mistakes and errors.

-        it is better to analyze the complexity of the algorithm in terms of required number of different operations.

-        more analysis is important to convince the readers - go through other literature

improve the readability of the manuscript in terms of typos mistakes and errors.

Round 2

Reviewer 4 Report

No further comments.